# Multiparametric MRI for Staging of Prostate Cancer: A Multicentric Analysis of Predictive Factors to Improve Identification of Extracapsular Extension before Radical Prostatectomy

**DOI:** 10.3390/cancers14163966

**Published:** 2022-08-17

**Authors:** Marina Triquell, Lucas Regis, Mathias Winkler, Nicolás Valdés, Mercè Cuadras, Ana Celma, Jacques Planas, Juan Morote, Enrique Trilla

**Affiliations:** 1Department of Urology, Vall d’Hebron University Hospital, 08035 Barcelona, Spain; 2Department of Surgery, Univesitat Autònoma de Barcelona, 08193 Barcelona, Spain; 3Imperial Urology, Charing Cross Hospital, Imperial Urology, Imperial College Healthcare NHS, London W6 8RF, UK

**Keywords:** prostate cancer, tumor staging, multiparametric magnetic resonance, extracapsular extension

## Abstract

**Simple Summary:**

In this multicentric study, we tested the accuracy of multiparametric magnetic resonance imaging (mpMRI) in detecting extracapsular extension (ECE) out of the prostate in order to plan surgical sparing of neurovascular bundles in radical prostatectomy. Univariate and multivariate logistic regression analyses were performed to identify other risk factors for ECE. We found that it has a good ability to exclude extracapsular extension but a poor ability to identify it correctly. Risk factors other than mpMRI that predicted ECE were as follows: prostatic specific antigen, digital rectal examination, ratio of positive cores, and biopsy grade group. We suggest that using mpMRI exclusively should not be recommended to decide on surgical approaches.

**Abstract:**

The correct identification of extracapsular extension (ECE) of prostate cancer (PCa) on multiparametric magnetic resonance imaging (mpMRI) is crucial for surgeons in order to plan the nerve-sparing approach in radical prostatectomy. Nerve-sparing strategies allow for better outcomes in preserving erectile function and urinary continence, notwithstanding this can be penalized with worse oncologic results. The aim of this study was to assess the ability of preoperative mpMRI to predict ECE in the final prostatic specimen (PS) and identify other possible preoperative predictive factors of ECE as a secondary end-point. We investigated a database of two high-volume hospitals to identify men who underwent a prostate biopsy with a pre-biopsy mpMRI and a subsequent RP. The sensitivity, specificity, positive predictive value (PPV), and negative predictive value (NPV) of mpMRI in predicting ECE were calculated. A univariate analysis was performed to find the association between image staging and pathological staging. A multivariate logistic regression was performed to investigate other preoperative predictive factors. A total of 1147 patients were selected, and 203 out of the 1147 (17.7%) patients were classified as ECE according to the mpMRI. ECE was reported by pathologists in 279 out of the 1147 PS (24.3%). The PPV was 0.58, the NPV was 0.72, the sensitivity was 0.32, and the specificity was 0.88. The multivariate analysis found that PSA (OR 1.057, C.I. 95%, 1.016–1.100, *p* = 0.006), digital rectal examination (OR 0.567, C.I. 95%, 0.417–0.770, *p* = 0.0001), ratio of positive cores (OR 9.687, C.I. 95%, 3.744–25.006, *p* = 0.0001), and biopsy grade in prostate biopsy (OR 1.394, C.I. 95%, 1.025–1.612, *p* = 0.0001) were independent factors of ECE. The mpMRI has a great ability to exclude ECE, notwithstanding that low sensitivity is still an important limitation of the technique.

## 1. Introduction

A classical selection of men at risk of prostate cancer (PCa) is achieved by the determination of serum prostate-specific antigen (PSA), digital rectal examination (DRE), and the recent incorporation of multiparametric magnetic resonance imaging (mpMRI) to combine systematic ultrasound (US)-guided biopsies with targeted biopsies [1].

Once PCa is detected, risk stratification must be performed to assure the correct radical treatment selection in localized diseases. The assessment of multiple clinical and laboratory parameters can be useful to predict pathological staging, biochemical recurrence, clinical progression, and cancer-specific survival [2]. Given its widespread use in PCa diagnosis, mpMRI has been proposed as a feasible tool for PCa local staging before radical prostatectomy (RP) [3].

The presence of extraprostatic extension (EPE) in the prostate specimen, which includes extracapsular extension (ECE) and seminal vesicles invasion (SVI), has been related to a significantly increased risk of progression and cancer-specific mortality [4]. The correct identification of ECE on mpMRI is crucial in order to plan nerve-sparing surgery to preserve erectile function, notwithstanding this can be penalized with worse oncologic results [5]. Magnetic resonance imaging, together with the classical clinical parameters, has demonstrated a central role in improving the ability to detect adverse pathological features in prostatic specimens [6,7]. However, this technique has been reported to have high specificity but low sensitivity for the detection of ECE, SVI, and overall stage [8].

The aim of this study was to assess the ability of preoperative mpMRI to predict ECE in the final prostatic specimen. The secondary end-point was to identify preoperative predictive factors to improve the identification of ECE in the final prostatic specimen.

## 2. Materials and Methods

### 2.1. Study Population and Intervention

We investigated the permanent RP database of two tertiary high-volume hospitals in the United Kingdom and Spain. Data from 1263 consecutive men, who underwent RP between January 2014 and September 2020 due to PCa, were enrolled in this study. The inclusion criteria were patients with an mpMRI prior to a prostate biopsy. Patients with incomplete data were excluded.

In this cohort, all patients underwent previous prostate biopsy due to PCa suspicion through an abnormal DRE or elevated PSA. The prostate biopsy protocols were different in each institution, but in both mpMRI-US fusion targeted biopsies of suspicious lesions were performed in the case of PIRADS (Prostate Imaging Reporting and Data System) ≥ 3. For each prostate cancer-affected core in the biopsy, the Gleason grade group was reported based on the International Society of Uropathology (ISUP) 2014 consensus [9]. The RP techniques varied according to the physician criteria as follows: open, laparoscopic, or robotic-assisted due to center preferences and case selection. Neurovascular bundle preservation in RP was carried out according to the proposal of Tewari et al. [10], and extended pelvic lymph node dissection was performed when the lymph node involvement risk was higher than 5% in the Memorial Sloan Kettering Cancer Center nomogram [11].

### 2.2. MRI Measurements

All patients underwent an mpMRI using a pelvic phased-array coil in the Siemens Magnetom Trio (3T) and Avanto (1.5T) platform, according to the European Society of Urogenital Radiology’s recommendations [12]. The high-resolution turbo spin-echo T2-weighted images (T2-WI) consisted of a 3-mm slice thickness and a small field of view imaging (180 mm). The DWI sequence initially consisted of the following 3b values: 50, 400, and 800. The patients received intravenous gadolinium contrast at 0.1 mmol/kg, injected at a rate of 2 mL/s. The DCE sequences consisted of a gradient recall echo, which was imaged over 4 min, with a matrix of 168,256 and a 3.3-mm slice thickness. The MRI-derived prostate volume (PV) was calculated from the ellipsoid formula as follows: 0.52 × (D1 × D2 × D3) [13]. Expert radiologists in each institution analyzed the images and reported the PV and image staging, as well as the presence, size, location, number, and malignant likelihood, according to the PI-RADS grading of suspicious lesions. The radiologists who read the images were blind to the objective of the study at the moment of the image report.

### 2.3. Population Analysis

The preoperative variables reported were as follows: age, DRE, resonance image staging, PSA, free PSA, prostate volume measured in mpMRI, PSA density (PSAD), ratio of affected cores, number of cores affected by PCa, maximum percentage affected by PCa in a core, and biopsy Gleason grade group (GG) according to the ISUP classification [14]. The definition of a positive core is a core affected by any length or percentage and of any grade of prostate cancer (ISUP GG ≥ 1). The ratio of the affected cores was considered the ratio between the cores affected by any grade of PCa and all cores obtained from the biopsy. The postoperative variables were as follows: final specimen pathological stage and ISUP GG. We considered the confirmed ECE as pathological stage T3a (pT3a). The results from the mpMRI were compared to the RP specimens.

### 2.4. Statistical Analysis

A descriptive analysis of the preoperative and postoperative variables was reported. The mean of the continuous variables and the absolute or relative frequencies for the categorical variables were calculated. The sensitivity, specificity, positive predictive value, and negative predictive value of the mpMRI in predicting ECE were assessed. A univariate analysis was performed to find the association between image staging and pathological staging. A multivariate analysis and a logistic regression were performed to find the association between the clinical parameters and the presence of ECE. A *p*-value below 0.05 was considered significant. The statistical analysis was performed using the IBM SPSS v27 Statistics Base.

## 3. Results

A total of 1147 patients fulfilled the selection criteria. The demographics and clinical information are listed in Table 1. The mean age was 62.57 years old. The mean PSA was 9.45 ng/mL. The most prostate biopsies were reported as ISUP GG 2 (455 out of 1147, 39.7%), followed by 322 reported as GG 1 (28.1%), 240 as GG 3 (20.9%), 86 as GG 4 (7.5%), and 44 as GG 5 (3.8%). The mean total biopsy cores obtained in the biopsies were 14.54 (1–73), and the mean number of positive cores was 5.07 (1–23). A total of 953 out of the 1147 (83.08%) patients underwent mpMRI by a 3T MRI machine. In this cohort, the most common image staging was iT2a (697 out of 1147, 60.8%). A total of 203 (17.7%) patients were classified as iT3a (ECE), according to image staging reported in the mpMRI, and 45 (3.9%) of them were defined as iT3b (SVI).

The pathological features of the RP pieces are recorded in Table 2. The most common pathological GG was 3 (510 out of 1147 patients, 44.5%), followed by 221 cases of GG 2 (19.3%). The pathological stage was pT2c in 501 RP specimens (43.7%); pT3a (ECE) was reported in 279 patients (24.3%) and pT3b (SVI) in 110 (9.6%). A total of 118 out of the 279 (42.29%) patients affected by ECE had a capsular invasion suspicion highlighted in the MRI.

The univariate analyses showed that MRI staging was statistically correlated with pathological staging, *p* < 0.05. The magnetic resonance imaging findings were compared to the final pathology. The MRI positive predictive value (PPV) of the mpMRI to detect ECE in the final pathological specimen was 0.58, and the negative predictive value (NPV) was 0.72. In order to detect capsular disease, the magnetic resonance sensitivity was 0.32, and the specificity was 0.88.

Multivariate analysis and logistic regression were performed, including the following preoperative variables: PSA, PSAD, DRE, maximum affected core, ratio of positive cores, and biopsy ISUP GG. The results are shown in Table 3. The independent predictors of ECE in the final specimen were as follows: PSA (OR 1.057, C.I. 95%, 1.016–1.100, *p* = 0.006), DRE (OR 0.567, C.I. 95%, 0.417–0.770, *p* = 0.0001), ratio of positive cores (OR 9.687, C.I. 95%, 3.744–25.006, *p* = 0.0001), and biopsy ISUP GG (OR 1.394, C.I. 95%, 1.025–1.612, *p* = 0.0001). Overall, PSAD was not a predictor of ECE. In order to better identify which patients with ECE suspicion in the preoperative MRI have a higher probability to indeed present capsular invasion in the final specimen, we established different clinical parameter cut-offs. A PSAD ≥ 0.20 ng/dL/cc was an accurate threshold to predict pT3awith a good discrimination performance (AUC = 0.701).

## 4. Discussion

RP, regardless of the open, laparoscopic, or robotic approach, is set as the standard of care for PCa in low-, intermediate-, and high-risk disease, with the same oncologic results as external beam radiotherapy [15]. The main adverse effects derived from surgery are erectile dysfunction and urinary incontinence, which can be observed in up to 74.7% and 21.3% of patients 12 months after surgery, respectively [16]. These complications imply a high impact on men’s quality of life [17]. The introduction of RP with a nerve-sparing technique has improved erectile function and urinary continence outcomes in the treatment of localized PCa [18].

Currently, the clinical stage of PCa before surgery is based only on DRE [19]. A DRE that suggests extension outside of the prostate capsule is classified as a high-risk disease, according to d’Amico’s classification groups for biochemical recurrence of PCa [20]. According to the current evidence, when there is a risk of ipsilateral ECE, nerve-sparing surgery is not recommended [21]. However, DRE’s sensitivity to detect a clinical stage ≥ T3a compared to mpMRI has been found to be low (12% vs. 51%, *p* < 0.001) [22]. Considering this scenario, mpMRI has been proposed as a tool for assessing the presence of ECE before surgery.

Until now, radiologists have been focusing on high specificity readings to reduce the unnecessary exclusion of men for curative treatment while maintaining the lowest false-positive results for ECE [23]. However, in daily clinical practice, a high sensitivity and a negative predictive value would be useful to draw up a better surgical plan. It would help to reduce the positive surgical margins for better oncologic outcomes, especially in intermediate- and high-risk patients [24]. Furthermore, a better capsular invasion suspicion would contribute to improving the selection of men for neuro-vascular bundle preservation. This better accuracy would be reflected in better functional results, particularly in low-risk patients [23].

According to the current recommendations, mpMRI should be performed before prostate biopsies to perform fusion targeted biopsies on suspicious lesions [25]. However, recent reports highlight its low implementation in everyday clinical practice [26]. Even though the role of mpMRI for disease local staging remains unclear [24], the implementation of mpMRI for staging has a good rate of acceptance among the urologist community according to recent surveys [27,28,29]. Moreover, mpMRI has been rated as a reliable tool for correctly identifying local staging [30].

De Rooij and colleagues [24] published a meta-analysis of 526 patients showing a pooled specificity of 0.9 (95% C.I., 0.88–0.93) and a pooled sensitivity of 0.57 (95% C.I., 0.49–0.65), but high differences in the sensibility results were reported between the studies. Our study shows a similar specificity (0.88), suggesting that the mpMRI can precisely exclude ECE. On the other hand, a low sensitivity (0.32) is reported in the current study aligned with known published data [31,32]. The lower sensibility in the present paper might be caused by the heterogeneity in the mpMRI technique, which includes 1.5T and 3T devices without an endorectal coil.

When a T2WI and an additional functional technique (DWI or DCE) are used, an improvement in the sensitivity to identify capsular invasions compared to T2WI alone has been reported [24]. In the current study, a combination of those three functional techniques was performed. Moreover, a high magnetic field strength device (3T) may also improve sensitivity [33].

The use of an endorectal coil allows for better special and spectral resolution, which should increase the accuracy of local staging; nonetheless, a meta-analysis shows there is no significant difference in the local staging performance between an endorectal coil and external modalities [34]. Cerantola et al. show a diagnostic accuracy of 62% for the endorectal coil-MRI in detecting ECE with a low sensitivity (35%) and a high specificity (90%). Based on these results, the authors suggest that this imaging should not be used as a first-line test to assess local invasions [35].

As in most of the published data, the radiologists in the current study reported the presence of a locally advanced disease as a dichotomic description as follows: “yes or no”. It has been suggested that a 5-point standardized lexicon of diagnostic certainty—from “unlikely” to “consistent with” ECE—may reduce the number of expressions used by radiologists to indicate their levels of diagnostic help to improve accuracy [36].

The risk of adverse pathology has been estimated classically using a combination of preoperative variables, such as Partin tables, which include PSA, Gleason score, and clinical stage; likewise, the Memorial Sloan Kettering Cancer Centre (MSKCC) nomogram also incorporates the age and percentage of positive cores in the biopsies [37]. Other parameters, such as PSAD, have been reported to be independent predictors of advanced pathological features in high-risk patients [38]. The use of modern markers, such as the Prostate Health Index (PHI) and PCA3 [39], PHI [40], 4K [41], and the Stockholm-3 test [42], have also been proposed to better predict adverse pathologies. The addition of mpMRI information to clinical parameters can improve the accuracy of detecting ECE [43,44,45]. Nevertheless, these nomograms were based on patients undergoing systematic prostatic biopsies and their application is questioned in the targeted biopsy era [46].

In line with the published studies, our study shows that mpMRI alone is not good enough to predict local staging. Gandaglia and colleagues proposed novel risk models based on the population who underwent MRI-targeted biopsies. Their results showed that models including both mpMRI data and the percentage of clinically significant PCa in systematic biopsies yielded the highest discrimination for ECE (AUC: 73%; 95% C.I.: 67–75%) [7]. Nonetheless, external validation was developed, showing no differences in the discrimination between Gandaglia’s mpMRI-based novel calculator and the MSKCC nomogram (AUC 71.8% vs. 69.8%, *p* = 0.3). Indeed, a minimal net benefit was highlighted [47].

The main limitation of the current study is its retrospective design. Nonetheless, all of the assessed images were informed by a radiologist prior to RP, which eliminates a potential selection bias. The fact that the reported biopsy does not differentiate between positive systematic and targeted biopsies means we cannot estimate the weight of the mpMRI-targeted positive core rate alone in the prediction of ECE, which can also be considered a limitation. The main strengths of our study are the high number of patients included from two different institutions and the high level of specialized radiologists and pathologists.

## 5. Conclusions

The specificity and NPV of the mpMRI are reasonable to decide on a surgical approach, notwithstanding that a low sensitivity reading is still an important limitation of the technique. Therefore, the benefit of mpMRI alone to predict ECE remains unclear and it should not be used as a screening tool alone. It seems useful to consider other clinical parameters, such as PSAD, to better predict capsular disease. The development and external validation of risk calculators, including mpMRI and other parameters, may lead to a higher discrimination in detecting ECE and could be a feasible tool for decision-making.

## Figures and Tables

**Table 1 cancers-14-03966-t001:** Clinical and pathological preoperative characteristics (N = 1147).

**Mean Age, Years (Range, SD)**	62.57 (81.9–50.9, 8.9)
**Mean tPSA, ng/mL (Range, SD)**	9.45 (0.7–80, 6.9)
**Mean % FreePSA (Range, SD)**	1.07 (0–10, 0.7)
**Mean PSAD, ng/mL^2^ (Range, SD)**	0.24 (0.21–2.29, 0.2)
**Mean PV, mL (Range, SD)**	44.72 (13–170, 19.9)
**No. Abnormal DRE (%)**	446 (38.9)
**No. Biopsy ISUP GG (%)**	
ISUP 1	322 (28.1)
ISUP 2	455 (39.7)
ISUP 3	240 (20.9)
ISUP 4	86 (7.5)
ISUP 5	44 (3.8)
**Mean Ratio Affected Cores (SD)**	0.37 (0.25)
**Mean No. Positive Cores (Range, SD)**	5.07 (1–23, 3.3)
**Mean Max. Affected Core (%) (Range, SD)**	43.56 (10–100, 28.8)
**No. 3T MRI (%)**	953 (83.08)
**No. Stage at mpMRI (%)**	
No Lesion	22 (1.9)
iT2a	697 (60.8)
iT2b	83 (7.2)
iT2c	96 (8.4)
iT3a (ECE)	203 (17.7)
iT3b (SVI)	45 (3.9)

Abbreviations: tPSA, total prostate specific antigen; No., number; PSAD, prostate specific antigen density; PV, prostate volume; DRE, digital rectal examination; ISUP, International Society of Urological Pathology; GG, grade group; ECE, extracapsular extension; SVI, seminal vesicle invasion; mpMRI, multiparametric magnetic resonance imaging; SD, standard deviation.

**Table 2 cancers-14-03966-t002:** Pathological postoperative characteristics (N = 1147).

**No. Pathological ISUP GG (%)**	
ISUP 1	136 (11.9)
ISUP 2	221 (19.3)
ISUP 3	510 (44.5)
ISUP 4	220 (19.2)
ISUP 5	58 (5.1)
**No. Pathological Stage (%)**	
pT2a	191 (16.6%)
pT2b	57 (5%)
pT2c	501 (43.7%)
pT3a (ECE)	279 (24.3%)
pT3b (SVI)	110 (9.6%)
pT4	9 (0.8%)
**No. of Confirmed ECE**	118

Abbreviations: ISUP, International Society of Urological Pathology; GG, grade group; ECE, extracapsular extension; SVI, seminal vesicles invasion.

**Table 3 cancers-14-03966-t003:** Multivariate analyses of preoperative variables for predicting ECE.

	Odds Ratio (CI 95%)	*p* Value
**PSA**	1.057 (1.016–1.100)	0.006
**PSAD**	1.719 (0.254–8.338)	0.501
**PSAD Cut-Off 0.2 ng/mL^2^**	0.627 (0.4–0.984)	0.042
**PSAD Cut-Off 0.15 ng/mL^2^**	0.966 (0.558–1.586)	0.89
**PSAD Cut-Off 0.10 ng/mL^2^**	0.785 (0.44–1.402)	0.414
**Abnormal DRE**	0.567 (0.417–0.770)	0.0001
**Max. Affected Core**	0.597 (0.259–1.375)	0.225
**Ratio Positive Cores**	9.687 (3.744–25.006)	0.0001
**Biopsy ISUP GG**	1.394 (1.025–1.612)	0.0001

Abbreviations: CI, confidence interval; PSA, prostate specific antigen; PSAD, prostate specific antigen density; DRE, digital rectal examination; ISUP, International Society of Urological Pathology; GG, grade group scenario; mpMRI has been proposed as a tool for assessing the presence of ECE before surgery.

## Data Availability

The data can be shared up on request.

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
