# Peer review of "Multiparametric MRI for Staging of Prostate Cancer: A Multicentric Analysis of Predictive Factors to Improve Identification of Extracapsular Extension before Radical Prostatectomy"

_cancers, 2022, doi:10.3390/cancers14163966_

Round 1

Reviewer 1 Report

Thank you for the opportunity to review the manuscript entitled, "Multiparametric MRI for staging of prostate cancer: a multicentric analysis of predictive factors to improve identification of extracapsular extension before radical prostatectomy." The authors present an analysis of 1147 men who underwent radical prostatectomy at a hospital in the UK or Spain between January 2014 and September 2020. The inclusion criterion was patients who underwent preoperative mpMRI. They evaluate the ability of mpMRI to predict extracapsular extension (ECE) on the postoperative specimen. The authors found that 24.3% had ECE on the final specimen. The positive predictive value was 0.58, the negative predictive value was 0.72, the sensitivity was 0.32, and the specificity was 0.88. PSA (OR 1.1, 95%CI 1.0 – 1.1), digital rectal exam (OR 0.6, 95%CI 0.4 – 0.8), the ratio of positive cores (unsure what this means) (OR 9.7 95%CI 3.7 – 25.0), biopsy Gleason grade (OR 1.4, 95%CI 1.0 – 1.6) were independent predictors of EPE. The authors concluded that mpMRI has a remarkable ability to exclude ECE, that the low sensitivity is a limitation, and that different clinical parameters such as PSA helped discriminate between patients with ECE and those without. The clinical topic is important, as early detection of extracapsular extension in prostate cancer could be relevant for precision medicine treatment decision making and planning nerve-sparing surgery in prostate cancer, the most common cancer in men. However, I have several comments to improve the quality of the manuscript.

1. Were the radiologists who read the imaging aware that all subjects were part of this study and were scheduled for prostatectomy? If so, this should be included as it may impact how the studies are interpreted. 

2. Were all MRIs re-read as part of this study with the knowledge that was scheduled for RP, or was some reads done clinically as part of SOC? Were the MRIs done pre-biopsy or all pre-RP only?

3. Please change "sensibility" to "sensitivity" throughout the paper.

4. The authors mention they utilize the ratio of positive scores as a predictor of ECE in the logistic regression analysis, but the description does not make sense. It is a ratio of what? Is it the ratio of positive cores to all cores? 

5. What is the definition of a positive core? Any cancer? I would be curious if the authors have information on a per-core basis on clinically significant cancer (e.g., any percent cancer for cores with Gleason 3+4 and above)

6. It would benefit the paper if the authors could showcase in a figure a representative example, either only mpMRI or mpMRI + histology.

7. Please report standard deviations when reporting means.

Additional minor comments:

- It would be helpful to add a sentence to the background in the abstract and in the introduction describing why using mpMRI for ECE is a clinical need (i.e., result from nomograms too poor?). 

I think the part about how the findings impact a surgeon's plan for nerve-sparing needs to be clarified.

- In the abstract, could you please specify that the regression analysis is logistic regression

Reviewer 2 Report

Authors should be congratulated for their work. The topic is interesting: The correct identification of extracapsular extension (ECE) of prostate cancer (PCa) on multiparametric magnetic resonance image (mpMRI) is crucial in order to plan the nerve-sparing approach in radical prostatectomy. 
The manuscript is well-written and easily readable, the methodology is robust and the tables are clearNevertheless, I suggest a major revision to improve the quality of the manuscript. Specifically, I suggest to the Author these manuscript (PMID: 34572950) and (PMID: 33348956), to improve the discussion section.

Author Response

Dear Reviewer 2,

We would like to thank you and the other reviewers for your comments. These have greatly  helped us to improve the quality of our manuscript.

Find the document attached, 

Reviewer 3 Report

Detailed information on MRI is the most important factor in this study.

At the very least, it is necessary to specify the ratio of whether the MRI of the target patient was performed at 1.5T or 3.0T with body coil or endorectal coil.

 We believe that detailed information about MRI, whether done at least at 3T or 1.5T, is needed to bring out the goodness of this research.

In the results in Table 1, the number of significant digits does not match.

Author Response

Dear Reviewer 3,

We would like to thank you and the other reviewers for your comments. These have greatly  helped us to improve the quality of our manuscript.

Find the new version of the manuscript attached. 

Review

Response

At the very least, it is necessary to specify the ratio of whether the MRI of the target patient was performed at 1.5T or 3.0T with body coil or endorectal coil.

Patients were undergone mpMRI using a pelvic phased-array coil.
We will write more details at the next version of the manuscript

 We believe that detailed information about MRI, whether done at least at 3T or 1.5T, is needed to bring out the goodness of this research.

We include it at the next version of the manuscript (Table 1).

In the results in Table 1, the number of significant digits does not match.

We will correct it at the next version of the manuscript

Round 2

Reviewer 2 Report

Authors should be congratulated for their work. The topic is interesting: The correct identification of extracapsular extension (ECE) of prostate cancer (PCa) on multiparametric magnetic resonance image (mpMRI) is crucial in order to plan the nerve-sparing approach in radical prostatectomy. 
The manuscript is well-written and easily readable, the methodology is robust and the tables are clearNevertheless, I suggest a major revision to improve the quality of the manuscript.

Reviewer 3 Report

The paper is well revised.